# Follicular Fluid Proteomic Analysis of Women Undergoing Assisted Reproduction Suggests That Apolipoprotein A1 Is a Potential Fertility Marker

**DOI:** 10.3390/ijms25010486

**Published:** 2023-12-29

**Authors:** Csilla Kurdi, János Schmidt, Zoltán Horváth-Szalai, Péter Mauchart, Krisztina Gödöny, Ákos Várnagy, Gábor L. Kovács, Tamás Kőszegi

**Affiliations:** 1János Szentágothai Research Center, University of Pécs, 7624 Pécs, Hungary; kurdi.csilla@pte.hu (C.K.);; 2Department of Laboratory Medicine, Medical School, University of Pécs, 7624 Pécs, Hungary; 3National Laboratory on Human Reproduction, University of Pécs, 7624 Pécs, Hungaryvarnagy.akos@pte.hu (Á.V.); 4Department of Biochemistry and Medical Chemistry, Medical School, University of Pécs, 7624 Pécs, Hungary; 5Department of Obstetrics and Gynecology, Medical School, University of Pécs, 7624 Pécs, Hungary; 6MTA-PTE Human Reproduction Scientific Research Group, University of Pécs, 7624 Pécs, Hungary

**Keywords:** assisted reproductive treatment, follicular fluid, proteomics, mass spectrometry, apolipoprotein A1, HDL cholesterol, predictive value

## Abstract

Infertility affects millions worldwide, posing a significant global health challenge. The proteomic analysis of follicular fluid provides a comprehensive view of the complex molecular landscape within ovarian follicles, offering valuable information on the factors influencing oocyte development and on the overall reproductive health. The follicular fluid is derived from the plasma and contains various proteins that can have different roles in oocyte health and infertility, and this fluid is a critical microenvironment for the developing oocytes as well. Using the high-performance liquid chromatography-mass spectrometry method, we investigated the protein composition of the follicular fluid, and after classification, we carried out relative quantification of the identified proteins in the pregnant (P) and non-pregnant (NP) groups. Based on the protein–protein interaction analysis, albumin and apolipoprotein A1 (ApoA1) were found to be hub proteins, and the quantitative comparison of the P and NP groups resulted in a significantly lower concentration of ApoA1 and high-density lipoprotein cholesterol in the P group. As both molecules are involved in the cholesterol transport, we also investigated their role in the development of oocytes and in the prediction of fertility.

## 1. Introduction

Infertility is described as a condition where a clinical pregnancy cannot be achieved after 12 months of regular, unprotected sexual intercourse. According to the WHO definition, infertility is a disease that causes disability by impairing normal reproductive functions [1]. Infertility affects 8–12% of reproductive-aged couples worldwide [2]. Since infertility affects millions of individuals, understanding the factors contributing to this disease is of paramount importance in guiding effective fertility treatments and interventions. The follicular fluid is a crucial component of the female reproductive system, and therefore, its analysis can provide valuable information on the microenvironment where the oocyte development occurs. By studying the composition of the follicular fluid, it is possible to find key biomarkers and factors that influence fertility offering a hope for improved diagnostic methods and personalized therapies for individuals struggling with infertility [3].

Despite the increasing use of assisted reproduction technologies (ARTs) during the last decades, the effectiveness of these procedures still remains under the desired level. To help pinpointing several proteins that could serve as potential biomarkers or treatment targets for achieving successful ART, pregnancy-related proteomics has been extensively used [4]. Proteomics is a type of ‘omics’ focusing on the examination of the interactions, functions, composition and structures of proteins and their cellular activities [5]. The progress in mass spectral (MS) profiling has empowered the identification of potential biomarkers and treatment targets in biological fluids linked to human reproduction. This advancement offers crucial information on different stages of the ART process and raises hopes for successful outcomes in high-risk ART pregnancies [4].

The follicular fluid (FF) serves as a natural environment for the oocyte, encompassing a complex network of proteins that undergoes dynamic changes. These proteins play an important role in the oocyte’s growth and development, significantly influencing its quality and potential for fertilization. Fortunately, FF is readily accessible as it is abundantly collected during ART procedures at the oocyte retrieval stage. It has been demonstrated to be a valuable substrate for proteomic analysis, facilitating the exploration of its precise composition and functions [4]. FF is a complex biological fluid found within the follicular antrum and provides a microenvironment for the developing oocytes and contains hormones, enzymes, anticoagulants, reactive oxygen species and antioxidants. It also plays a crucial role in facilitating communication among cells within the antral follicle and serves as a vital medium delivering nutrients to the oocyte [6]. The composition of follicular fluid is derived partially from the plasma by diffusion and transudation, and the wall of the follicle synthetizes metabolites that are going to be altered by granulosa and theca cells. This unique biological matrix is the only one directly linked to the oocyte, serving as the microenvironment for its growth and differentiation in vivo [7]. FF, like human serum, contains a wide array of proteins, each presented at varying concentrations [8]. The primary role of these proteins is to modulate oocyte and follicular maturation [9] and to protect the developing oocytes and other follicular cells from physical or oxidative damage. They play a crucial role in facilitating communication between somatic and germ cells, which is essential for the oocyte to acquire its competence [10]. Any changes or any disruptions in the follicular fluid microenvironment can result in altered protein function with a consequence of altered viability of the oocytes, thus causing adverse pregnancy outcomes [11]. A better understanding of how different molecular changes may interfere with fertility and pregnancy rates in assisted reproduction technologies would be of utmost importance and may provide new information for future clinical practice [12].

In the present work, we planned to investigate the protein composition of the FF, because proteomic analysis of FF provides a good tool to gather information on the follicular development and on the oocyte quality as well [13], providing a huge impact on the success of the ART. After identification of the FF proteins, we performed an analysis on their protein–protein interactions. Then, the most potent proteins were quantified, and the literature was surveyed to support the role of our best candidates and their potential significance on the quality of oocytes and on fertility as well.

## 2. Results

### 2.1. Demographic and Clinical Data of Patients

In Table 1, the demographic and clinical features of the patients enrolled in our study are shown.

### 2.2. Classification of the Proteins in the Follicular Fluids

A total of 131 individual proteins were identified in the FF samples. These proteins can be classified based on their role in biological processes, their biological classes, and according to the pathways they are involved in. The classification is illustrated in Figure 1. Based on the biological processes, the identified proteins were classified as follows: (1) *cellular processes* (20.7%), (2) *metabolic processes* (16.7%) and (3) *biological regulation* (14.9%). Other features such as *response to stimulus*, *localization*, *immune system processes*, *multicellular organism related processes*, *signaling* and *growth* pathways were also suggested. The proteins of the follicular fluid can be differentiated into different protein classes. The three biggest classes were the *defense/immunity proteins* (32.4%), the *protein binding activity modulator proteins* (27.9%) and the *transfer/carrier proteins* (16.2%). *Protein modifying enzyme proteins*, *metabolite interconversion enzyme proteins*, *transmembrane signal receptor proteins and intercellular signal molecules* classes were classified with lower percentages. Based on the pathways, the identified proteins involved in two main classes were distinguished. Pathways related to *blood coagulation* (44.4%) and the *plasminogen activating cascade* were the most important ones, but others such as *B cell activation*, *angiogenesis*, *integrin signaling pathways*, pathways related to *vitamin D metabolism* and *serine*–*glycine biosynthesis* were also presented.

### 2.3. Relative Quantification of the Identified Proteins

To evaluate our results more precisely, relative quantification was used to compare the intensities of the MS signals in the pregnant (P, n = 15) and non-pregnant (NP, n = 15) groups. In total, 77 proteins were found to have different levels in the two groups. Combining intensity values with protein–protein interaction and functional analysis can reveal the roles of proteins in specific biological functions or pathways. Proteins with both string interactions (altered intensity values) and relevant functions can be particularly interesting.

### 2.4. Protein–Protein Interactions (PPIs)

The purpose of the protein–protein interaction analysis was to monitor the physical connection between identified proteins and their interactions to understand how they interact with each other and carry out various biological processes. These interactions are crucial for the functioning of biological pathways, signal transduction and the regulation of gene expression. PPI analysis can help to uncover the complex network of interactions of cellular functions and can provide insights into disease mechanisms. By adding intensity values to the general PPIs results, it can provide additional quantitative information on the interactions, so for each protein, P/NP intensity value ratio was calculated and illustrated as a full STRING network. The results of the PPI analysis can be seen in Figure 2. The explanation of the abbreviations is listed in Appendix A.

In Figure 2, a full STRING network of our follicular fluid samples is shown. In this analysis, every node represents a protein and edges in the PPI network illustrate the interactions between the proteins. These interactions can be physical (such as direct binding) or functional, indicating proteins that cooperate in the same biological process. The layout of the nodes is also an important factor when analyzing the results. The arrangement of the nodes provides information on the clusters or groups of interacting proteins. Nodes close together often have more interactions among them. Based on the P/NP intensity value ratio calculation, there were some proteins that had a value above one; these were ALB, APOL1, A2M, APOC3, FGA, FGB, TTR, SERPIND1, GSN and ITIH1, meaning that they had higher intensity values in the P group. SERPINC1, AGT, ApoA1, AMBP, ORM1, HPX, SHBG, LBP and ITIH2 had lower ratio values, meaning that the intensity value of these proteins was higher in the NP group.

Proteins with many connections are called hub proteins, which are quite often crucial in the network. These hub proteins play important roles in biological systems and are usually involved in multiple interactions, making them crucial for maintaining the network’s structure and function, and these proteins may be targeted for further investigations as they are likely to be involved in critical biological functions. Node degree is the number of connections or interactions that a certain protein has with other proteins in the network. In Table 2, the node degrees of proteins in our network are shown. In our study, ALB had a high node degree meaning that this protein interacted with 27 other proteins, making it a hub protein. Another hub protein was ApoA1 that interacted with 18 other proteins.

### 2.5. Functional Enrichment Analysis

Following the PPI analysis, the nodes were associated with biological functions and pathways through enrichment analysis. Functional enrichment analysis is a commonly used method to identify trends in datasets and to predict functions and the involved pathways of the proteins of interest. The result of our analysis is shown in Table 3. The *Count in network* numbers show how many proteins in our network are annotated with a particular term, and the second number indicates how many proteins in total have this term assigned. The significance of the enrichment results was represented in two important statistical measures, namely, *Strength* and *False Discovery Rate* (FDR). *Strength* quantifies how much a certain functional term or pathway is overrepresented or enriched in the dataset. A higher value indicates a stronger enrichment signal, and it suggests that proteins associated with that term are significantly overrepresented in the dataset. This value can be used to rank the enriched terms. *FDR* is a statistical value, and it estimates the proportion of false positives among the statistically significant results. The threshold is typically set to 0.05, and it determines which enriched terms or pathways are significant after multiple testing corrections. Values below 0.05 are considered as significant.

In our dataset, the most significant terms were high-density lipoprotein (HDL) receptor binding, Phosphatidylcholine-sterol O-acyltransferase activator activity (Lecithin Cholesterol Acyltransferase [LCAT] activator activity) and Apolipoprotein receptor binding. These terms had the highest strength score, and their FDR values were also below 0.05, suggesting a high level of confidence in the association between these proteins and the term presented.

### 2.6. Quantitative and Statistical Analysis of the Selected Proteins

Based on the PPI and enrichment analysis, we concluded that there are two proteins that should be further quantified. These proteins were ALB and ApoA1. ALB was confirmed to be a central protein in the FF based on the PPI. ApoA1 was also found to be a central protein, and the functional enrichment analysis has proven that it is indeed a potentially interesting molecule in the FF. Since ApoA1 is a major component of the HDL molecule, the concentration of HDL cholesterol (HDL-C) in the follicular fluid was also determined. Haptoglobin (HP) was also investigated because it was found to be associated with ApoA1 in the FF. Previous research indicated that HP has a significant influence on fertility [14]. Another study showed that HP in the human follicular fluid inhibits the reverse transport of cholesterol to the bloodstream by blocking the apolipoprotein A1-induced activation of LCAT [15]. The results of the quantitative measurements are shown in Table 4.

#### 2.6.1. Comparison of the Protein Concentrations Based on the Outcome

Based on the outcome, two groups were separated, and pregnant (n = 15) and non-pregnant (n = 15) groups were determined. There were no significant differences in the concentration of TP and ALB concentration of the P and NP groups. In the P group, the concentration of the ApoA1 and HDL-C was lower, and *p* values were 0.007 and 0.010, respectively, showing significant differences between the P and NP groups. The HDL/ApoA1 ratio was also significantly altered between the two groups (*p* = 0.045). The concentration of HP and the HP/ApoA1 ratio did not show any significant differences (Table 4 and Table 5).

#### 2.6.2. Comparison of Protein Concentrations Based on the Age of the Patients

In comparison, two age groups were determined. One group (younger, n = 12) contained adult patients aged equal to or under 34, and the members of the other group were 35 years old or older. In this comparison, none of the proteins showed significantly altered levels (Table 5).

#### 2.6.3. Comparison of the Protein Concentrations Based on the BMI

The patients were divided into two groups according to their BMI. Patients with BMI values between 18.5 and 24.9 were considered to belong to the normal group (n = 13), while in the overweighted group, the BMI score was above 25 (n = 16). There were no significant differences between the normal and overweight groups (Table 5).

## 3. Discussion

In this article, we investigated the composition of the FF and characterized its content based on PPI and enrichment network analysis. In many proteomic studies, during the sample preparation, the most abundant proteins such as albumin, different subclasses of immunoglobulins, haptoglobin, transferrin, fibrinogen and ApoA1 are removed because their signals tend to obscure the signal of the less abundant proteins. Their high abundance interferes with the detection of other proteins and can prevent accurate detection. By depleting them, the quality and depth of the analysis can be greatly enhanced, but unfortunately, removing these components may result in the loss of precise detection of other follicular fluid proteins with a much lower concentration. Therefore, we did not remove the abundant proteins to save all of the protein contents in the FF. After analysis of the follicular fluid, we concluded that there are several proteins that can be involved in the lipid metabolism and transport in the FF. Quantitative measurements were made to determine the concentrations of those proteins that may have an important role in the fertility and in the outcome of ART. To investigate the effects of the age and the BMI of the patients on the protein pattern of the FF, statistical analysis was performed. ApoA1 showed a significantly lower concentration in the FF of the pregnant patients compared with the NP individuals, and the same was found for HDL-C as well. Both molecules are involved in the transport of cholesterol, which is a vital molecule in early mammalian development. It is essential for membrane fluidity, making it a crucial component for the synthesis of cell membranes in rapidly dividing cells [16]. The metabolism of cholesterol is an important aspect of producing good-quality embryos, supported by the finding that ApoA1 is produced in the embryo because of the rapid cell division and the intense membrane synthesis [17]. Cholesterol serves as a precursor for steroid hormones, playing a pivotal role in the maturation of ovarian follicles [18]. In the follicular stage, the thecal cells absorb cholesterol from blood HDL and produce androgens (androstenedione and testosterone). These substrates cross the basal membrane and stimulate estrogen production by granulosa cells within the developing follicle. Following the LH surge, the follicular basal membrane becomes more permeable to larger lipoproteins, increasing cholesterol availability to support active production during ovulation [19]. Following ovulation, the follicle becomes vascularized, and the follicular cells utilize stored cholesterol esters, while increasing the expression of lipoprotein receptors. In the corpus luteum, the luteal cells employ cholesterol from both de novo biosynthesis and plasma LDL and HDL to significantly boost progesterone synthesis [20].

In our research, we compared the concentration of ApoA1 in the FF of the pregnant and the non-pregnant patient groups, and we observed that the concentration of ApoA1 was significantly lower in the pregnant group. ApoA1 is a protein that is part of the lipoprotein class. It is a crucial protein for cholesterol metabolism, and it also helps in the synthesis of HDL molecules. ApoA1 makes up 70% of the HDL molecules being produced in the liver (80%) and in the gut (20%) as well [21]. Some studies reported that ApoA1 has a significant role in female reproduction and in the positive outcome of pregnancies [22]. Patients with different diseases were examined, and it was revealed that there is an elevated level of ApoA1 that can be associated in preeclampsia, endometriosis, PCOS and repeated implantation failure. A recent study identified elevated ApoA1 levels in women who experienced spontaneous preterm delivery [23]. To achieve a successful pregnancy, multiple other factors should be taken into consideration. There were several reports suggesting the association of ApoA1 with the quality of the embryos, their development and viability [24]. Several past studies have linked the levels of ApoA1 to embryo quality in IVF settings associated with the success of IVF trials [25]. Regarding the expression of the ApoA1 protein in the culture medium of day 2 to 3 embryos, it was determined that ApoA1 was less abundant in the samples resulting in pregnancy [26]. Also, based on previous research, it was observed that there is a connection between successful embryo transfer and low levels of ApoA1 in the early culture medium, suggesting that the internalization of ApoA1 might be a characteristic feature of viable embryos. This idea was supported by the presence of ApoA1 mRNA transcripts in day 5 blastocysts but not in day 3 embryos, indicating higher ApoA1 concentrations inside the blastocysts [17]. A recent study on bovine embryos revealed a decrease in proteins related to lipid metabolism, including a 1.6-fold reduction in ApoA1 abundance from the oocyte stage to the four-cell stage. This decline could be attributed to an increased secretion ability of the embryo into the culture medium during development [27].

The other protein that was present in a lower concentration in the pregnant group was HDL-C. HDL is the smallest and most compact class of lipoproteins [28]. The follicle is permeable only to proteins with a molecular weight of up to 300 kDa, and therefore, serum molecules up to this size can be found in the FF. HDL is the sole lipoprotein that can be detected there [29]. The HDL in the FF is primarily sourced from the plasma and compared to the plasma, it varies in both size and composition, having lower cholesterol content and higher levels of phospholipids [30]. HDL-C may play a crucial role in fertility because it is a main class of lipoproteins present in significant quantities in the FF. It is believed that follicular HDL may transport essential lipids to follicular cells or facilitate the removal of cholesterol from these cells. A key process for delivering HDL-C to ovarian cells, especially those involved in steroid production, is known as selective lipid uptake, and it is facilitated by the HDL receptor SR-B1. Vital knowledge about the connection between HDL metabolism and female fertility has been gained from studies involving SR-B1 knock-out (KO) mice. These mice have high levels of cholesterol-rich HDL in their bloodstream leading to female infertility. Interestingly, this infertility can be reversed by administering cholesterol-reducing medication. Oocytes released during ovulation in SR-B1 KO females display dysfunction and accumulate excessive cholesterol. However, the precise mechanisms by which the FF HDL contributes to maintaining oocytes’ cholesterol balance are not yet understood [31].

In a review, it was suggested that FF HDL plays a role in maintaining balanced cholesterol levels in developing oocytes by removing excess cholesterol from the plasma membrane. If there is a disruption in the reverse cholesterol transport process caused by abnormal FF HDL, there is an accumulation of unesterified cholesterol in the egg. This imbalance can interfere with crucial processes such as oocyte maturation, meiosis arrest and egg activation following fertilization. Consequently, disruptions in HDL-C metabolism could impair FF HDL function, leading to infertility due to dysfunctional and unstable eggs [32].

The outcome of ART can be unfavorably affected by a woman’s increasing age. The declines in fertility—generally, the likelihood of a successful pregnancy—among individuals begin at around ages from 31 to 35. By the age of 40, at least half of the patients may experience infertility. The exact reasons behind women’s age-related decline in fertility are not fully understood. However, several factors may contribute to this phenomenon, including a reduction in the ovarian reserve, poorer quality of oocytes, lower rates of successful implantation, ovulatory issues stemming from changes in hormonal patterns and potential uterine problems [33]. Previous studies have also shown that the composition of follicular fluid changes with age, and these changes could be associated with the quality of eggs [11,34]. Both age and obesity are linked to systemic oxidative stress, and in this physiological context, elevated levels of specific lipids could potentially trigger harmful processes that negatively impact reproductive function [35]. It was previously presented that a higher BMI can be associated with fertility problems and adverse pregnancy outcomes [36]. Also, higher BMI and obesity are associated with changes in the composition of the FF including alterations in its lipid profile. Increased levels of triglycerides, cholesterol and inflammatory markers in the FF have been observed in obese individuals [37]. The BMI-dependent concentration of HDL-C in the serum was significantly associated with those of the FF concentrations, and this suggests that changes in serum cholesterol levels could influence the quality of the oocytes. Additionally, the article implied that the number of HDL particles in FF might adversely affect the developmental capacity of the oocyte, even if fertilization occurred normally. Notably, there is a proposition that the combined impact of FF HDL-C and ApoA1 has a protective role in oocyte health and embryo development by reducing embryo fragmentation [38]. Interestingly, our results did not show any significant alteration between the follicular fluid ApoA1 and HDL-C concentration of the younger/older groups and normal weight/obese groups.

The field of reproductive biology is quite complex and is under continuous exploration and discovery. Despite the interactions occurring within the ovarian microenvironment, there is a notable gap in our understanding of the potential influence of HDL and its major component, ApoA1, on the development and health of the oocyte and the pregnancy outcomes. Surprisingly, the scientific literature is relatively scarce in providing comprehensive information on the role of HDL and ApoA1 in these fundamental biological processes. This paucity of research is particularly striking given the critical functions that can be attributed to HDL and ApoA1 in lipid metabolism, inflammation modulation and antioxidant defense. These two molecules have been extensively studied in the context of cardiovascular health where their roles in cholesterol transport and cardiovascular risk reduction is well documented. However, their potential impact on female reproductive health, especially concerning oocyte maturation, fertilization and pregnancy outcomes remains an unexplored territory. Understanding the interplay between the HDL, ApoA1 and the ovarian microenvironment has importance not only for advancing our knowledge of reproductive biology but also for unrevealing potential therapeutic avenues to enhance fertility treatments and to improve pregnancy success rates. This knowledge gap could start a new research field that can contribute to the development of more effective interventions and personalized strategies for couples struggling with infertility.

The major strengths of this study could be the relatively controlled and homogeneous sample size, making it easier to draw comparisons between pregnant and non-pregnant groups. This study’s focus on follicular fluid samples provides a specific analysis, potentially offering more information into the local microenvironment and its effect on the developing oocytes and fertility. Our study has some limitations. Our analyses were performed on only a limited number of patients. The smaller sample size may limit the generalizability of the findings to a broader population, so the conclusions drawn should be further supported by studies on a much larger patient population. However, the highly sophisticated HPLC/MS technique is not widely accessible, and therefore, we think that our study is a useful start for a large-scale analysis of the FF in various laboratories worldwide.

When fully validated on a much larger sample size, the proteins analyzed in our study may prove to be clinically relevant for predicting the outcome of the ART. Protein, lipid and lipoprotein analysis, which are commonly available in clinical laboratories, could offer timely information before and during IVF treatment. We would like to emphasize that molecular studies—especially proteomics—of the FF in relationship with ART are still limited, which makes interpretation of the results a bit difficult.

## 4. Materials and Methods

### 4.1. Patient Enrollment

The study was conducted between May 2021 and March 2023 at the Department of Obstetrics and Gynecology (FF sampling) and at the National Laboratory on Human Reproduction (analytical studies), University of Pécs, Hungary. Detailed information was given to all patients or their next-of-kin regarding our study protocol, while written consent was obtained from all. Exclusion criteria were patients under 18 years of age, unobtainable or withdrawn consent and autoimmune diseases. Patients with endometriosis or polycystic ovary syndrome (PCOS) were not included in this study. The study protocol was approved by the Regional Research Ethics Committee of the University of Pécs (no. 5273-2/2012/EHR), conforming to the 7th revision of the Helsinki Declarations (2013). All of the 30 patients enrolled in this study received assisted reproductive treatment (ART). The inclusion criteria were either male infertility or female infertility caused by tubal problems. The obtained patient data were based on the medical records of the hospital informational system.

In our study, a total of 30 ART patients were involved, and 15 of these patients belonged to the pregnant and 15 to the non-pregnant groups. The pregnant group was defined as patients having biochemical pregnancy. Biochemical pregnancy was defined by an elevated hCG level (>5 IU/L) 14 days after the embryo transfer.

### 4.2. Follicular Fluid Sample Collection

The GnRh agonist triptorelin was used in both long and short protocols, and cetrorelix was applied in the antagonist protocols. Depending on the maturity of the follicles, individual doses of rFSH ranging from 150 to 250 IU per day were used for stimulation. The starting dose was determined based on BMI and age. A maximum daily dose of 300 IU was given to those with a previously determined low response. The stimulation was supplemented with rLH or human menopausal gonadotropin (hMG) individually, according to the patient’s age or response. From the 6th day of the cycle, the follicular maturity was monitored by ultrasound every consecutive day. Gonadotropin was administered individually according to the size of the follicles. When at least two follicles exceeded 17 mm in diameter, 250 µg (6500 IU) of recombinant human chorionic gonadotropin was given parenterally to induce final oocyte maturation. Aspiration was performed 36 h later using an ultrasound-guided transvaginal puncture under routine intravenous sedation. The follicular fluid collection was performed during oocyte retrieval procedure, and follicular fluid samples from individual follicles were pooled. After the procedure, the samples were centrifuged immediately at 6700× *g* for 10 min at room temperature to remove the erythrocytes, white blood cells and granulosa cells. The supernatant was collected and stored at −80 °C for further analyses.

### 4.3. Chemicals

To avoid protein loss during the sample preparation, Protein LoBind Eppendorf tubes (Eppendorf AG, Hamburg, Germany) were used. For the protein separation, ice-cold chloroform/methanol solution was applied (50/50, *v*/*v*, Merck Life Science, Darmstadt, Germany). After precipitation of the proteins, rehydration buffer (6 M urea, Bio-Rad Laboratories, Hercules, CA, USA; 100 mM TRIS, Bio-Rad Laboratories, Hercules, CA, USA) at pH 7.8 was used. For the determination of the protein concentration, micro-BCA (bicinchoninic acid) assay (ThermoFisher Scientific, Darmstadt, Germany) was utilized. The overnight enzymatic digestion was performed by Trypsin/Lys-c enzyme mixture (Promega Corporation, Madison, WI, USA). For sonication, Sonics and Materials Inc VCX 130 ultrasonic homogenizer (53 Church Hill Road, Newtown, CT, USA) was applied. Purification of the digested peptides was performed by an Oasis HLB 96-SPE well-plate (WAT058951; Waters, Milford, CT, USA) instrument. The eluted samples were concentrated with an Eppendorf Concentrator Plus system (Eppendorf AG, Hamburg, Germany). The purified peptides were redissolved in 100 µL of water containing 0.1% formic acid (Merck Life Science, Darmstadt, Germany). During the HPLC separation, Solvent A and Solvent B were used. Solvent A consisted of water/formic acid (99.9/0.1, *v*/*v*), (Merck Life Science, Darmstadt, Germany) while solvent B was composed of water/formic acid/acetonitrile (4.9/0.1/95, *v*/*v*/*v*), (Merck Life Science, Darmstadt, Germany).

#### 4.3.1. Sample Preparation for HPLC/MS Measurement

The thawed samples (300 µL) were transferred into Protein LoBind Eppendorf tubes and 1000 µL of ice-cold chloroform/methanol solution was added to each sample. After vortexing, the samples were centrifuged for 10 min at 10,000× *g* at 4 °C, and the separated protein pellets were collected into new tubes. The precipitated proteins were dried for 10 min using the Eppendorf Concentrator Plus system, and after that, the proteins were resuspended in a rehydration buffer supported by ultrasonication. Samples were individually sonicated on ice for 1 min using a 20% power setting on the micro-tip probe and a 50% pulse. The total protein concentrations were measured with a micro-BCA assay, and all samples were normalized to a protein mass of 150 µg before proceeding with the enzymatic digestion. The overnight in-solution enzymatic digestion was performed with a Trypsin/Lys-C enzyme mixture. Before digestion, the protein extracts were reduced and alkylated according to the manufacturer’s technical bulletin (V5071). The digested peptides were purified using an Oasis HLB 96-SPE well-plate, and the eluted samples were concentrated with an Eppendorf Concentrator Plus system. The purified peptides were redissolved in 100 µL of water containing 0.1% formic acid.

#### 4.3.2. Parameters of the HPLC/MS Method

Nano separation-based proteomic analysis was performed by Bruker EASY-nLC equipment coupled with Bruker Maxis 4G UHR-QTOF instrument (Bruker Daltonics, Bremen, Germany). Five μL aliquots of the samples were injected and separated by a homemade C18 analytical column (3 μm, 75 μm × 150 mm) using a 120 min long multistep gradient elution at a flow rate of 250 nL min^−1^. Two different solvents (Solvent A and B) were used for the separation. The mass spectrometer was operated in positive ion mode, and the scanning range was set to 300–2200 m/z. The flow rate of nebulizer gas was 2 L/min at a pressure of 0.6 bar, and the temperature was set to 160 °C. The capillary voltage was 4.5 kV, and the 30 most intensive peptide peaks were selected for collision-induced dissociation (CID) fragmentation.

### 4.4. Measurement of ApoA1, Haptoglobin and HDL Cholesterol

Apolipoprotein A1 and haptoglobin levels were measured by immunoturbidimetric assays on a Cobas Integra 400 Plus analyzer, while HDL cholesterol was determined by a homogeneous enzymatic colorimetric test on the c502/c702 modules of a Cobas 8000 analyzer (both instruments from Roche Diagnostics GmbH, Mannheim, Germany) at the Department of Laboratory Medicine (University of Pécs, Hungary; accreditation registration number: NAH-9-0008/2021).

### 4.5. Data Analysis

The obtained data from the MS measurements were processed using Bruker Data Analysis 4.4 software (Bruker Daltonik, Bremen, Germany). The protein identification was carried out using Mascot server (V2.4.1, Matrix Sciences Inc., Boston, MA, USA). Searching parameters were set to allow one missed cleavage site, accepting a 50 ppm mass error in MS1 and 0.3 Da in MS2 mode. The significance threshold was set to *p* ≤ 0.05 for reliable identification. The classification of the identified proteins was performed using PANTHER classification system (v.17.0, https://www.pantherdb.org/ (accessed on 18 April 2022)). The protein–protein interaction network analysis of the identified proteins was performed by using STRING database (v.12.0, https://string-db.org/ (accessed on 21 August 2023)). For relative quantification, MaxQuant software (v2.2.0.0, Max-Planck-Institute of Biochemistry, Munchen, Germany) was utilized in which a label-free quantification (LFQ) method was set. LFQ techniques rely on measuring the peak intensity or area under the curve of the identified peptide ions to enable the comparative quantification of peptides across various samples.

The statistical analysis of the obtained results was performed using SPSS (version 28.0.0.0, IBM SPSS Statistics, Armonk, New York, NY, USA) and OriginPro 2023b (OriginLab, Northampton, MA, USA) programs. To determine the differences between the groups, Mann–Whitney U-test was performed. Values of *p* < 0.05 were considered statistically significant.

## 5. Conclusions

In conclusion, our investigation regarding the composition of the follicular fluid of pregnant and non-pregnant patients has revealed lower concentrations of ApoA1 and HDL-C in the pregnant group. The observed alterations in ApoA1 and HDL-C levels suggest a potential association between lipid metabolism and infertility, opening new ways for further research into the underlying mechanisms especially because these two molecules’ potential impact on the female reproductive health is still a relatively unexplored field. As these lipid components play crucial roles in cellular function and health, understanding their specific impact on the follicular microenvironment could pave the way for targeted interventions and may improve the success of the ART.

## Figures and Tables

**Figure 1 ijms-25-00486-f001:**
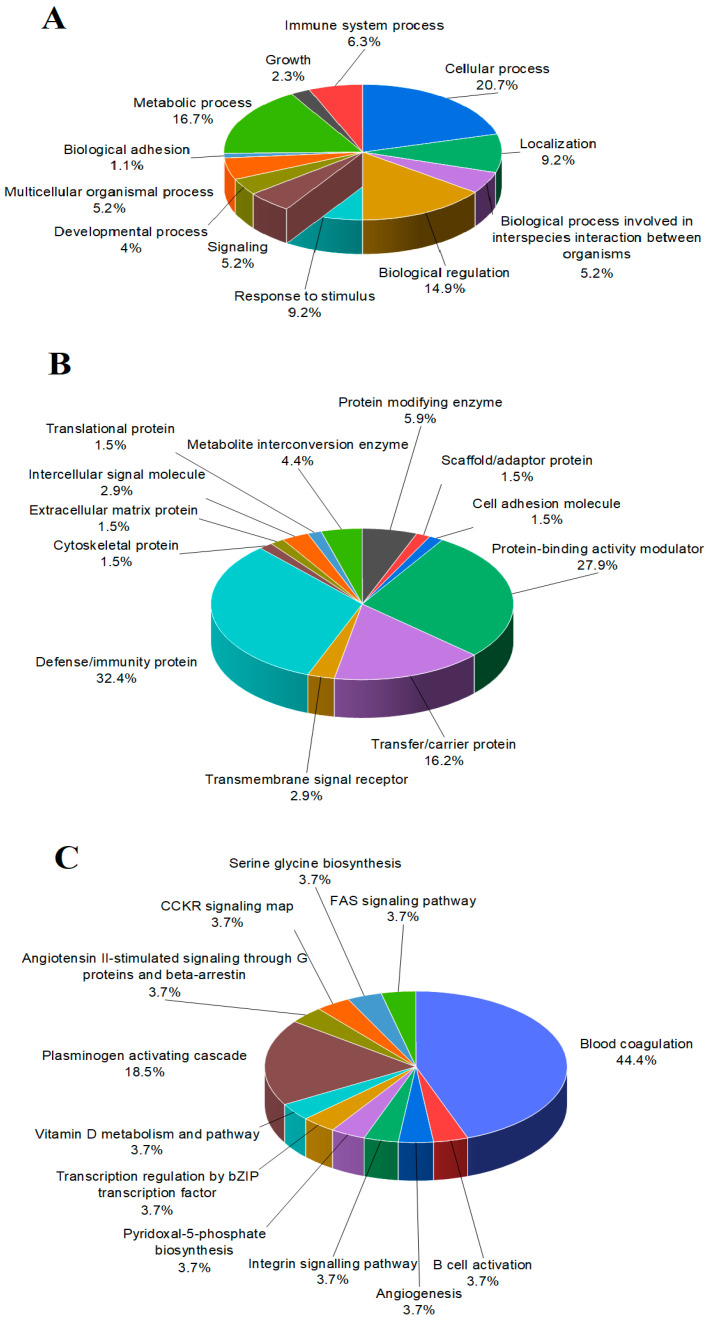
Gene ontology analysis of the proteins identified in human follicular fluid. Proteins were classified according to (**A**) biological processes, (**B**) protein classes and (**C**) pathways involved. Results are displayed as percentage of proteins classified to a category over the total number of class hits.

**Figure 2 ijms-25-00486-f002:**
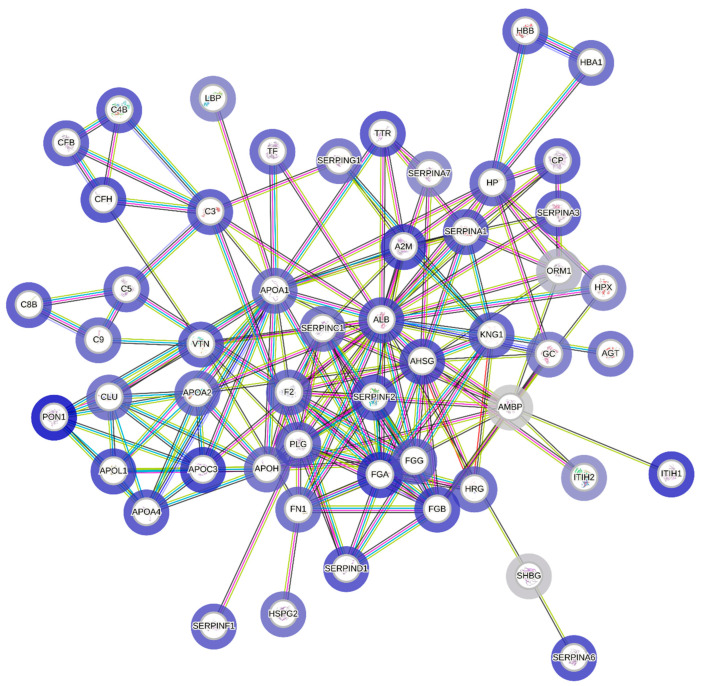
Protein–protein interaction network (full STRING network) of the identified proteins. The color of the lines illustrates the type of interaction evidence (e.g., turquoise = known interactions from curated databases, purple = experimentally determined, predicted interactions based on green = gene neighborhood, red = gene fusions, dark blue = gene co-occurrence, other associations such as yellow = text mining, black = co-expression and light blue = protein homology). Dark-blue color of the halo represents that the intensity values of the certain protein is higher than 1 P/NP ratio value, and light-blue color represents lower P/NP ratio values than 1. For the minimum required interaction score, 0.9 was set, which provides the highest confidence.

**Table 1 ijms-25-00486-t001:** Clinical characteristics of the patients involved in our research.

	Pregnant [Mean ± SD] (n = 15)	Non-Pregnant [Mean ± SD] (n = 15)
Age	36.31 ± 5.51	32.93 ± 4.94
BMI	26.31 ± 5.84	25.92 ± 5.65
Number of oocytes retrieved	12.27 ± 7.54	11.85 ± 8.4
Number of fertilized oocytes	4.86 ± 3.68	3.15 ± 2.69
Number of IVF cycles	1.89 ± 0.9	2.07 ± 0.75

**Table 2 ijms-25-00486-t002:** The node degree of proteins in our network. Only proteins with node degree over 10 are shown.

Name of Node (Protein)	Node Degree
ALB	27
ApoA1	18
AHSG	17
FGA	14
APOH	13
PLG	13
AMBP	12
F2	12
FGB	11
FGG	11
HP	10
A2M	10
SERPINC1	10

**Table 3 ijms-25-00486-t003:** Results of the functional enrichment analysis based on intensity values of the pregnant and non-pregnant patient groups. The table represents proteins that are over-represented in a large set of proteins.

Term Description (Function)	Count in Network	Strength	False Discovery Rate
High-density lipoprotein particle receptor binding	3 of 3	2.5	0.00018
Phosphatidylcholine-sterol O-acyltransferase activator activity	3 of 6	2.2	0.00067
Apolipoprotein receptor binding	2 of 6	2.02	0.0426
Lipoprotein particle receptor binding	5 of 30	1.72	2.56 × 10^−5^
Serine-type endopeptidase inhibitor activity	16 of 98	1.71	9.09 × 10^−20^
Cholesterol transfer activity	3 of 23	1.62	0.0156
Endopeptidase inhibitor activity	21 of 177	1.58	2.24 × 10^−23^
Complement binding	3 of 26	1.56	0.0185
Phosphatidylcholine binding	3 of 29	1.52	0.0233
Peptidase regulator activity	22 of 227	1.49	2.28 × 10^−23^
Proteoglycan binding	3 of 37	1.41	0.0426
Cholesterol binding	4 of 52	1.39	0.0067
Steroid binding	7 of 101	1.34	1.62 × 10^−5^
Enzyme inhibitor activity	23 of 396	1.27	4.86 × 10^−20^
Chaperone binding	6 of 108	1.25	0.00044
Molecular carrier activity	4 of 72	1.25	0.0181
Glycosaminoglycan binding	13 of 245	1.23	6.10 × 10^−10^
Heparin binding	9 of 173	1.22	2.34 × 10^−6^
Antioxidant activity	4 of 76	1.22	0.0205
Extracellular matrix structural constituent	6 of 131	1.16	0.0012
Sulfur compound binding	10 of 272	1.07	6.86 × 10^−6^
Protease binding	5 of 135	1.07	0.016
Enzyme regulator activity	28 of 1239	0.86	8.74 × 10^−15^
Lipid binding	15 of 796	0.78	8.08 × 10^−6^
Molecular function regulator activity	30 of 1960	0.69	1.11 × 10^−11^
Signaling receptor binding	23 of 1499	0.69	3.28 × 10^−8^
Protein binding	40 of 7242	0.24	0.0022

**Table 4 ijms-25-00486-t004:** Results of the quantitative measurements of the selected proteins.

	Pregnant [Mean ± SD] (n = 15)	Non-Pregnant [Mean ± SD] (n = 15)
Total protein (g/L)	45.37 ± 6.21	47.61 ± 11.14
Albumin (g/L)	32.72 ± 4.25	34.43 ± 8.05
ApoA1 (mg/dL)	123.54 ± 16.51	139.96 ± 22.27
HP (g/L)	0.55 ± 0.373	0.43 ± 0.24
HP/ApoA1 ratio	0.46 ± 0.29	0.32 ± 0.26
HDL-C (mmol/L)	0.55 ± 0.24	0.78 ± 0.23
HDL/ApoA1 ratio	0.17 ± 0.07	0.21 ± 0.04

**Table 5 ijms-25-00486-t005:** Results of the statistical analysis of measured FF proteins. The significance level was set as *p* < 0.05. Data significantly influencing the pregnancy outcome are highlighted in bold.

Protein Name	Parameter	*p* Value
TP	Outcome	0.595
ALB	Outcome	0.567
HP	Outcome	0.367
ApoA1	Outcome	**0.007**
HDL-C	Outcome	**0.010**
HDL-C/ApoA1 ratio	Outcome	**0.045**
HP/ApoA1 ratio	Outcome	0.148
TP	Age	0.232
ALB	Age	0.267
HP	Age	0.439
ApoA1	Age	0.325
HDL	Age	0.325
TP	BMI	0.846
ALB	BMI	0.559
HP	BMI	0.449
ApoA1	BMI	0.268
HDL	BMI	0.503

## Data Availability

The data that support the findings of this study are available from the corresponding author upon reasonable request.

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
