# Peer review of "Follicular Fluid Proteomic Analysis of Women Undergoing Assisted Reproduction Suggests That Apolipoprotein A1 Is a Potential Fertility Marker"

_ijms, 2023, doi:10.3390/ijms25010486_

Round 1

Reviewer 1 Report

Comments and Suggestions for Authors

In this study the authors used HPLC/MS method to assess the protein composition of follicular fluid. They found ALB and ApoA1 to be hub proteins and that ApoA1 and HDL-cholesterol had a lower concentration in the pregnant group compared to the non-pregnant group.

The manuscript is clearly written, and results are well described and highly interesting. However, the method section is missing out on a lot of information to make it more understandable for the reader. I recommend the following changes, that will probably further improve the manuscript:

Material and methods:

·         Medications for stimulation and ovulation trigger are missing.

·         Information on number of mature oocytes is missing.

·         Information on embryo culture is missing.

·         Information on luteal phase support is missing.  

·         Please state when hCG test and clinical pregnancy were performed.

·         Please define the “pregnant group”. Biochemical pregnancy, clinical pregnancy, live birth?

·         Have there been abortions in the pregnant group? This might be of relevance for the results!

·         Was follicular fluid pooled or analyzed separately?

Results:

·         Proteins have been assigned to groups (Fi.1). What if proteins had more than one role in biological processes (for example cellular process & biological regulation)?

Discussion:

·         The analysis of FF needs to be clarified. If follicular fluid was analyzed separately, it would be of even more interest to know which embryo had which follicular fluid composition. Relate the respective FF composition to embryo quality and overall treatment outcome.

·         The analysis was based on outcome (pregnant versus non-pregnant). If FF was analyzed separately, why didn’t you correlate the FF composition to fertilization rate or embryo quality.

·         Add strengths and limitations of the study.

·       

Comments on the Quality of English Language

  A few typos exit within the manuscript. I would highly recommend having the manuscript proof-read once again.

Reviewer 2 Report

Comments and Suggestions for Authors

This is an interesting and well-written article consistent with the topics of the journal. 

My main concern is regarding the experimental design and how the authors assigned their 2 study groups. I would expect an explanation of the charasteristics of the populations included. For example, does the pregnant group consist of women who became pregnant after that particular IVF cycle when the samples were collected? What about women who became pregnant in a later cycle? It also seems that not all patients underwent the same number of cycles, so can this affect the results? The distinction between the groups is not clear. 

Other minor comments: 

Abstract: - You should avoid abbreviations like HPLC/MS, ALB...

- I would advise the authors to include a mention of the number of patients included and definition of the P/NP groups.

- A conclusion sentence would also be useful there.

Table 1: - Was a statistical analysis performed to detect potential differences in these parameters? 

L99: - Were all detected proteins common between the 2 groups? No protein expressed in just one of them?

L406: - In my opinion, the conclusion is not very clear. I would propose to end the article with a more specific conclusion of how the findings will contibute to the clinical management of infertile patients. The potential weaknesses of the study would be better fit in a separate paragraph before the conclusion.

Comments on the Quality of English Language

The manuscript is well-written in terms of the English language.

Reviewer 3 Report

Comments and Suggestions for Authors

In this manuscript, the authors are investigating the protein composition in follicular fluid in an attempt to find a correlation with pregnancy.

It is a potentially interesting paper; however, it can be improved.

I have some concerns:

1-are the follicular fluids obtained from single patients pooled or are analyzed separately?

2-on line 442-443 FF were centrifuged also to remove granulosa cells

3- the discussion is very long and not easy to understand. It must be rewritten.

4-lines 286-288. Does the altered expression mean that ApoA1 increased or decreased? In all the paragraphs, it is not easy to understand in what cases you find an increase and when you find a decrease.

5-In the discussion, I could not find mention of the paper by Tiffany Von Wald et al. (Fertility and Sterility Volume 93, Issue 7, 1 May 2010, Pages 2354-2361) where the authors describe a decline of ApoA1 in fertility with aging.

5- line 255 analyzation must be analysis

6- line 271. androstenedione and testosterone are substrates for estrogen production.

7- line 326-327 The follicle is permeable….therefore, serum molecules up to 300 kDa can be found in the FF.

8- in the paragraph starting on line 343, it is better to clarify the terminology: developing oocyte and eggs.

Comments on the Quality of English Language

No significant comments for the English. 

Round 2

Reviewer 2 Report

Comments and Suggestions for Authors

Congratulations to the authors for working on improving the clarity of their manuscript. In my opinion, the article is appropriate for publication.

Comments on the Quality of English Language

The English language of the text is good and only minor corrections may be needed.